# CFR-PEEK Pedicle Screw Instrumentation for Spinal Neoplasms: A Single Center Experience on Safety and Efficacy

**DOI:** 10.3390/cancers14215275

**Published:** 2022-10-27

**Authors:** Ann-Kathrin Joerger, Sebastian Seitz, Nicole Lange, Amir K. Aftahy, Arthur Wagner, Yu-Mi Ryang, Denise Bernhardt, Stephanie E. Combs, Maria Wostrack, Jens Gempt, Bernhard Meyer

**Affiliations:** 1Department of Neurosurgery, Klinikum Rechts der Isar, Technical University, 81675 Munich, Germany; 2Department of Neurosurgery, Helios Klinikum Berlin-Buch, 13125 Berlin-Buch, Germany; 3Department of Radiooncology, Klinikum Rechts der Isar, Technical University, 81675 Munich, Germany

**Keywords:** carbon reinforced polyethyl-ether-ether-ketone, CFR-PEEK, CFRP, spinal metastases, spinal primary bone tumors, neuro-oncology

## Abstract

**Simple Summary:**

Advances in screening methods and new therapeutic strategies have lead to a continuous decline in cancer death rates, especially over the last ten years. As a consequence, the number of patients with spinal metastases is increasing. In modern oncological treatment surgery followed by postoperative radiotherapy for spinal metastases has gained a decisive role. For spinal stabilization, pedicle screws and rods are used. They used to be made of titanium or cobalt–chrome alloys. Recently, carbon-fiber-reinforced (CFR) polyethyl-ether-ether-ketone (PEEK) was introduced as a new material reducing artifacts on imaging and showing less perturbation effects on photon radiation. The aim of this study is to report on the safety and efficacy of CFR-PEEK pedicle screw systems for spinal neoplasms in a large cohort of consecutive patients. We could show that implant-related complications, such as intraoperative screw breakage and screw loosening, were rare. So, we conclude that CFR-PEEK is a safe and efficient alternative to titanium for oncological spinal instrumentation.

**Abstract:**

(1) Background: Surgery for spinal metastases has gained a decisive role in modern oncological treatment. Recently, carbon-fiber-reinforced (CFR) polyethyl-ether-ether-ketone (PEEK) pedicle screw systems were introduced, reducing artifacts on imaging and showing less perturbation effects on photon radiation. Preliminary clinical experience with CFR-PEEK implants for spinal metastases exists. The aim of this monocentric study is to report on the safety and efficacy of CFR-PEEK pedicle screw systems for spinal neoplasms in a large cohort of consecutive patients. (2) Methods: We retrospectively analyzed prospectively the collected data of consecutive patients being operated on from 1 August 2015 to 31 October 2021 using a CFR-PEEK pedicle screw system for posterior stabilization because of spinal metastases or primary bone tumors of the spine. (3) Results: We included 321 patients of a mean age of 65 ± 13 years. On average, 5 ± 2 levels were instrumented. Anterior reconstruction was performed in 121 (37.7%) patients. Intraoperative complications were documented in 30 (9.3%) patients. Revision surgery for postoperative complications was necessary in 55 (17.1%) patients. Implant-related complications, such as intraoperative screw breakage (3.4%) and screw loosening (2.2%), were rare. (4) Conclusions: CFR-PEEK is a safe and efficient alternative to titanium for oncological spinal instrumentation, with low complication and revision rates in routine use and with the advantage of its radiolucency.

## 1. Introduction

In 2005, a milestone study demonstrated a significant advantage of patients with spinal metastases (SM) treated with surgery followed by radiotherapy over patients treated with radiotherapy alone regarding their functional status [1]. Moreover, it was shown that spinal surgery, mainly spinal instrumentation with decompression, for metastatic epidural spinal cord compression (MESCC) significantly improves pain, neurologic function and health-related quality of life [2,3]. Consecutively, surgery for SM has gained a decisive role in modern interdisciplinary oncological treatment [4,5,6].

Advances in screening methods and new therapeutic options, in particular the implementation of targeted therapies, have lead to a continuous decline in cancer death rates, especially over the last ten years [7]. Due to improvements of the treatment protocols of many cancer types, above all, cancer of the hematopoietic and lymphoid system has become a “chronic disease” with a nearly normal life expectancy [7]. As a consequence, the number of patients with SM is increasing [8]. This leads to a change of mindset, calling for new therapeutic strategies when treating spinal tumors and metastases to guarantee a maximum of independent life for these patients. While SMs are the most frequent neoplasms of the spine, being found in about 15–35% of cancer patients [8,9] primary bone tumors of the spine are rare (less than 5% of spinal neoplasms) [10]. Treatment decisions are made interdisciplinary and individually, taking into account the patient’s symptoms, the degree of osteolytic instability, the degree of epidural compression, the radio- and chemosensitivity of the tumor and the overall prognosis [4,11].

So far, pedicle screws and rods made of titanium or cobalt–chrome alloys were used for spinal instrumentation. However, these highly absorbing materials lead to artifacts in computed tomography (CT) scans, hampering CT image-based radiotherapy planning [12]. Recently, carbon-fiber-reinforced (CFR) polyethyl-ether-ether-ketone (PEEK) pedicle screw systems were introduced. CFR-PEEK reduces artifacts on CT and magnetic resonance imaging (MRI) and shows less perturbation effects on photon radiation than titanium [13,14,15]; thus, radiotherapy can be planned and applied more precisely and follow-up images can be analyzed better with respect to tumor recurrence (Figure 1 and Figure 2). A cadaver study has demonstrated the biomechanical non-inferiority of CFR-PEEK pedicle screws compared to titanium [16]. We and other authors have previously reported our clinical experience with CFR-PEEK implant systems as a viable alternative for patients with spinal neoplasms [13,17,18,19,20]. However, a large safety and efficacy study of CFR-PEEK spinal stabilization systems does not exist so far. Therefore, we analyzed a consecutive series of 321 patients undergoing posterior stabilization for SM and primary bone tumors of the spine using CFR-PEEK pedicle screws.

## 2. Materials and Methods

### 2.1. Study Design

We retrospectively analyzed prospectively the collected data of consecutive patients being operated on using a CFR-PEEK pedicle screw system for posterior stabilization (Icotec, Altstätten, Switzerland) because of spinal metastases or primary bone tumors of the thoracic or lumbar spine. Pedicle screw placement was performed and navigated using an operating room-based sliding gantry CT (Brilliance CT Big Bore, Philipps, Amsterdam, The Netherlands), a mobile cone-beam CT (O-arm II, Medtronic, Minneapolis, MN, USA) [21] or a C-arm with 3-dimensional scanning (Arcadis Orbic, Siemens, München, Germany). Cement augmentation was used, depending on the quality of the cancellous bone. Indication of surgery was discussed in an interdisciplinary neurooncological board consisting of certified neurosurgeons, oncologists, radiooncologists and neuroradiologists. Aspects of spinal instability or deformity, epidural compression, the patient’s functional status, comorbidities and the oncological burden of the disease were evaluated. Reconstruction of the anterior column was performed when needed, depending on preoperative imaging, the degree of instability and systemic tumor burden, either in the same surgery or as a staged second surgery. For vertebral body replacement, either an expandable PEEK cage (XRL, DePuySynthes, Solothurn, Switzerland), an expandable CFR-PEEK cage (Kong, Icotec, Altstätten, Switzerland) or an expandable titanium alloy cage (Obelisc, Ulrich Medical, Ulm, Germany) was used. All the surgeries were performed by six senior surgeons.

### 2.2. Population

The data comprise anonymized records of patients operated on in the period from 1 August 2015 until 31 October 2021 in the Department of Neurosurgery of a tertiary care hospital. Baseline demographic data, the Karnofsky performance status scale (KPS), surgical details, complications and the outcome of patients were analyzed.

### 2.3. Ethical Agreement

The study was approved by the ethical committee of our university (reference number 96/19 S) and conducted in accordance with the Declaration of Helsinki.

### 2.4. Statistical Analysis

Descriptive statistics were calculated using Microsoft Excel 2011 (version 14.0.0) (Redmond, Washington, DC, USA). Figures were calculated by IBM SPSS statistics (version 28.0.1.0) (Armonk, New York, NY, USA).

## 3. Results

### 3.1. Demographic Background

We included 321 patients, 306 with SM and 15 with primary bone tumors of the spine, of a mean age of 65 ± 13 years (Table 1). Most patients were of a KPS of 80% or better (Table 1). The most frequent primary tumor site for metastatic patients was the prostate followed by the breast and non-small-cell lung cancer (NSCLC) (Table 1). Primary bone tumors were chordoma (five cases), aneurysmatic bone cyst (four cases), fibrous dysplasia (three cases), angiosarcoma (one case), cavernous haemangioma (one case) and osteosarcoma (one case). Symptoms of patients were, in most cases, pain without neurological impairment (Table 2).

### 3.2. Surgical Details

In the majority, posterior stabilization was performed in the thoracic spine, followed by the lumbar spine. On average, 5 ± 2 levels were instrumented. In 257 cases (80.1%), a standard open approach via midline skin incision was used, while in 64 cases (19.9%), pedicle screws were inserted minimally invasively; i.e., transmuscular (Table 3). Additional decompression was performed in 248 cases (77.3%). Cement augmentation of pedicle screws was used in 77 cases (24.0%). Anterior reconstruction was performed in 121 patients (37.7%) (Figure 3 and Figure 4). The mean blood loss was 1104 mL (± 1146 mL). Intraoperative red blood cell transfusion was necessary in 133 (41.4%) patients. For patients with primary bone tumors, in almost all cases (except of one palliative), an extensive tumorresection was performed. In eleven cases (73.3%), a total vertebrectomy with vertebral body replacement was performed. In five cases, the tumor was embolized preoperatively.

### 3.3. Complications and Revision Surgery

Intraoperative complications were documented in 30 out of 321 (9.3%) patients (Table 4). Direct implant-associated complications as screw breakage were rare (eleven out of 321 cases, 3.4%). In six cases, pedicle screws broke during insertion; in two cases, during intraoperative revision; and in three cases, during implant removal. The tips of the broken screws were left in the vertebral body. During insertion, in five cases, another pedicle screw was inserted in the same level in a different trajectory, and in three cases, the level was skipped on the side of the broken screw without the need of extension of the construct as all these screws broke in the middle part of the fusion.

Revision surgery for postoperative complications was necessary in 55 (17.1%) patients (Table 5, Figure 3). The revision rate because of pedicle screw loosening was low (seven out of 321 patients, 2.2%) (Table 5). The reasons for screw loosening were low-grade infection (three cases), acute putrid infection (two cases), mechanical screw pullout (one case) and tumor recurrence (one case). In one case, revision surgery was necessary because of rod breakage. However, this rod was made of titanium. In one case, which was revised multiple times because of an acute infection, a postoperative screw breakage was registered.

### 3.4. Outcome

In total, 258 (80.4%) patients were treated with radiotherapy postoperatively. Nine patients with spinal metastases needed reoperation due to local tumor recurrence. The median time to tumor recurrence was 417 days (range: 301–1261 days). For six patients with primary bone tumors, the first operation in our department was already a revision surgery because of a recurrent tumor with a median time interval of 389.5 days (range: 28–1836 days). One of these patients had another revision surgery because of tumor recurrence after 141 days, and another patient was operated twice (after 266 days and after another year). One patient with a first-time diagnosis of a spinal primary bone tumor was operated after 182 days for tumor recurrence. The median follow-up for all the patients was 97 days (range: 7–1888 days). Seven patients died during the same hospital stay. The reasons were respiratory insufficiency (five), cardiopulmonary decompensation (one) and palliative situation (one). The majority of patients preserved or even improved their neurological function postoperatively (Table 6). Analogously, for the majority, the postoperative KPS was equal or even better (Table 6).

## 4. Discussion

In this study, we report about the safety and efficacy of CFR-PEEK pedicle screw systems for patients with SM and primary bone tumors of the spine.

The rate of intraoperative complications of our study was comparable with other series of spinal instrumentation for spinal neoplasms [22]. The rate of intraoperative implant-associated complications was low. In eleven cases, screw breakage was reported: six during insertion, two during intraoperative revision and three when removing the implants. In four cases of screw breakage during insertion, an osteoblastic bone was documented and in two cases, the reason remained unclear. Biomechanical studies have shown that CFR-PEEK stabilization constructs resist the same static and cyclic axial compression loading and pull-out forces as titanium does [16,23]. However, torsion forces during screw insertion have not been analyzed in these studies. The rate of CFR-PEEK screw breakage in our study was comparable to what has been reported before in a smaller cohort of patients [18].

The major reasons of postoperative complications requiring revision surgery were surgical site infections and wound healing disorders, which were regarded not to be attributed to the use of CFR-PEEK. Their number was comparable to the rates of this type of complication of patients treated for SM previously reported by other studies [24,25]. In seven cases (2.2%), revision surgery for screw loosening was performed. This rate is lower compared to what has been described for titanium alloy systems (16%) [26,27]. It is also significantly lower compared to the rate of pedicle screw loosening after CFR-PEEK instrumentation for spondylodiscitis (35%), which we had examined in another study [28]. The mean time of the diagnosis of screw loosening in the latter study was 110 days, while the median follow-up for patients with spinal metastases in this study was 79 days and for patients with a primary bone tumor, 349 days. So, the shorter follow-up interval of patients with spinal metastases in this study could bias the rate. However, pedicle screw loosening after spinal instrumentation for infectious indication and for oncological indication cannot be compared directly because of different factors influencing the loosening process, such as the use of cement augmentation, different bone quality, the role of biofilm-producing bacteria and different surgical strategies.

The preoperative KPS of patients in our study was within the range of other recent studies about surgical treatment for spinal neoplasms [24,29]. A strong association between KPS and survival after surgery for spinal metastases was shown before [30]. In our study, the majority of patients preserved or even improved their KPS and neurological function postoperatively. Modern oncological therapeutic concepts and better screening methods have led to a prolonged survival of cancer patients [7]. As a consequence, modern spinal tumor surgery not only aims on spinal stabilization, prevention of or recovery from neurological deficits and pain reduction, but also on long-term symptom control. Therefore, durable constructs are required, enabling an optimal application of adjuvant radiotherapy and an optimal long-term follow-up imaging. These requirements become even more clear given the fact that the majority of patients in our study presented with pain without neurological impairment, while the percentage of patients with neurological symptoms was higher in older studies [31].

It has been shown in vitro [32] and in vivo [13] that CFR-PEEK reduces artifacts on CT and MR imaging and shows less perturbation effects on radiotherapy dose distributions [15] than titanium, fulfilling the requirements for an optimal application of radiotherapy (Figure 5 and Figure 6) and optimal long-term follow-up imaging. The advantages of CFR-PEEK on follow-up imaging have already been shown in the field of pyogenic spondylodiscitis [33].

### 4.1. Strenghts of This Study

This is the largest study, to the best of our knowledge, of consecutive patients with spinal neoplasms operated on using routinely a CFR-PEEK pedicle screw system reporting on safety and efficacy.

### 4.2. Limitations of This Study

There are several limitations of this study. (1) It is a retrospective study without a randomized control group; thus, CFR-PEEK cannot be compared to titanium directly. (2) No standardized follow-up examinations were performed and the follow-up period was comparably short. (3) In reconstructing the anterior column in some cases, titanium cages were used as well as in some cases, titanium rods for posterior stabilization, degrading the advantages of CFR-PEEK pedicle screws regarding artifacts. However, this study did not aim on a qualitative evaluation of postoperative imaging and radiotherapy planning.

## 5. Conclusions

CFR-PEEK is a safe and efficient alternative to titanium for spinal instrumentation because of spinal neoplasms with low complication and revision rates in routine use. We recommend using CFR-PEEK for spinal oncological surgery in the context of modern cancer therapy so that patients can benefit from an optimized application of radiotherapy and from an earlier detection of tumor recurrence. To further prove these obvious clinical advantages, prospective studies with long-term follow-up are necessary.

## Figures and Tables

**Figure 1 cancers-14-05275-f001:**
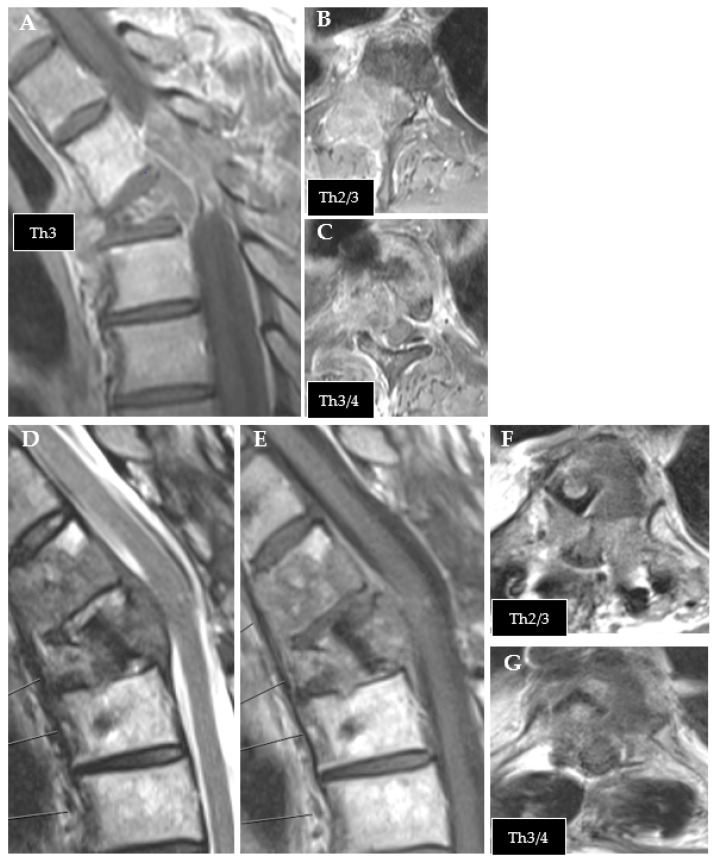
Exemplary case of tumor recurrence. (**A**–**C**): preoperative MRI of thoracic metastasis from urothelial carcinoma (T1 contrast enhanced). (**D**–**G**): follow-up MRI after CFR-PEEK instrumentation showing tumor recurrence ((**D**): T2; (**E**–**G**): T1 contrast enhanced).

**Figure 2 cancers-14-05275-f002:**
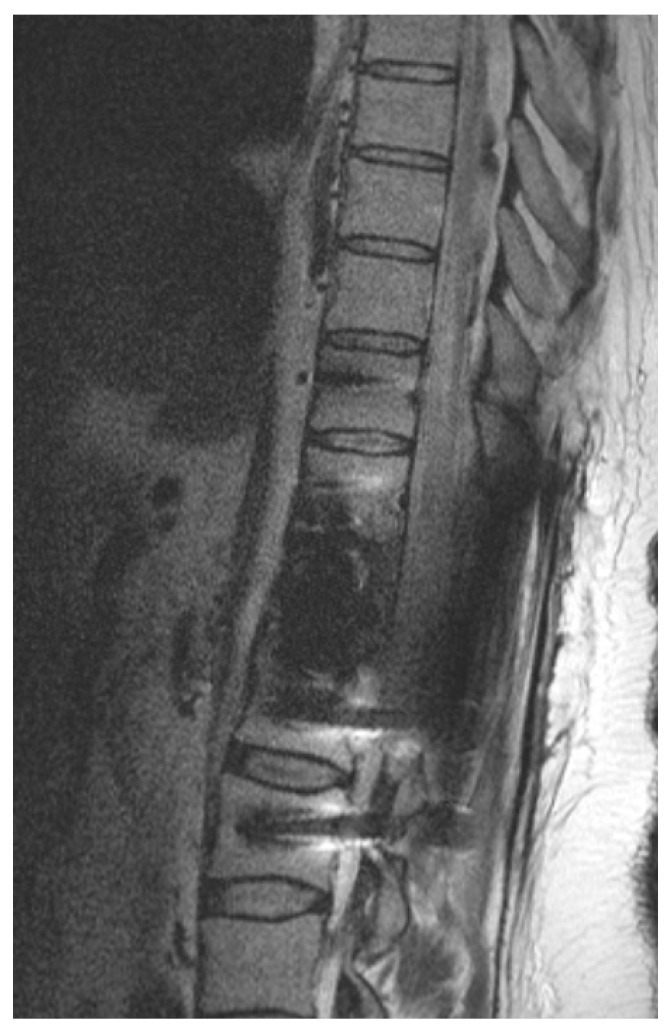
Illustrative example of postoperative imaging. Postoperative 1.5 T MRI after CFR-PEEK instrumentation and resection of renal cell carcinoma metastasis that was embolized preoperatively. Patient showed paraplegia postoperatively. Because of reduced artifacts, MRI could rule out epidural hematoma and depicted spinal cord edema caused by ischemia.

**Figure 3 cancers-14-05275-f003:**
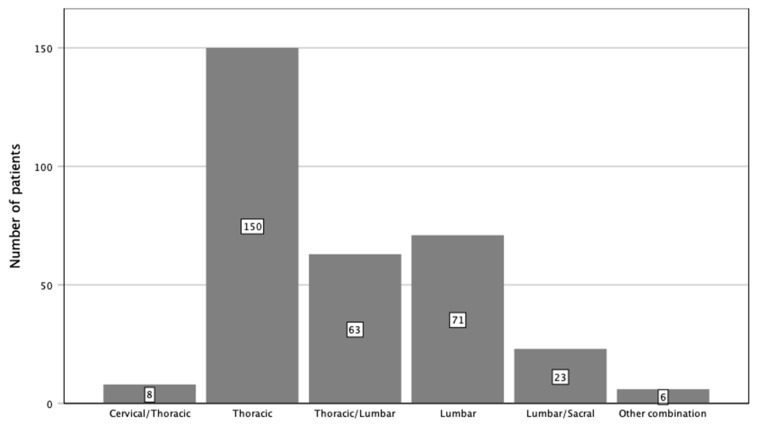
Localization of instrumentation stratified by spinal sections.

**Figure 4 cancers-14-05275-f004:**
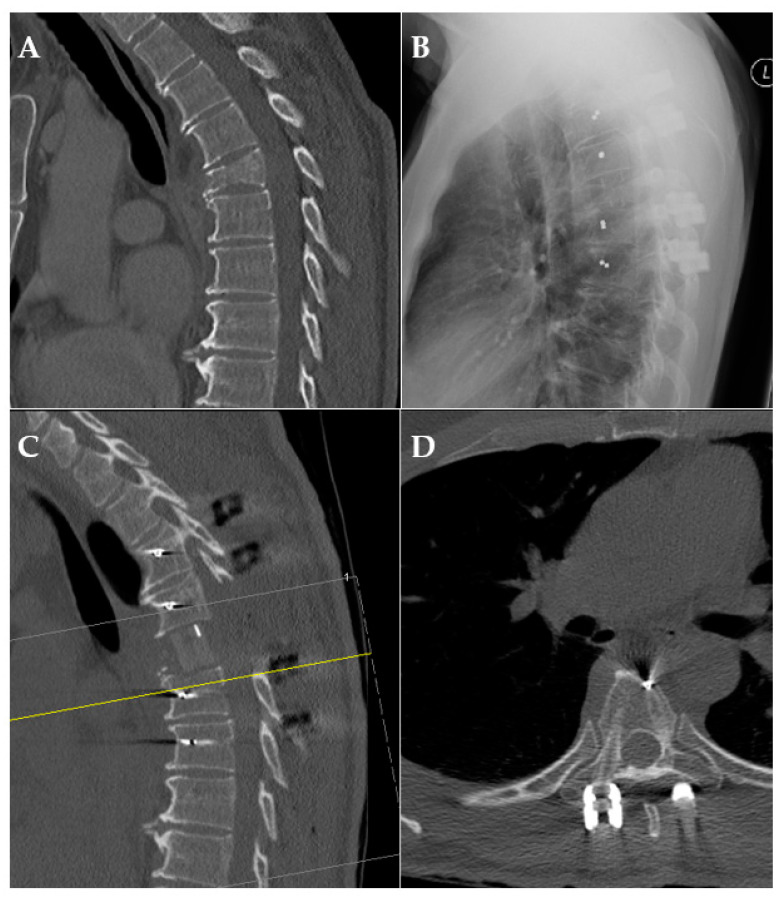
Exemplary case of pre-/postoperative CT scan and X-ray. (**A**): preoperative CT scan of a pathological fracture of the thoracic spine; (**B**): X-ray after posterior fusion; (**C**,**D**): CT scan after posterior fusion and anterior reconstruction demonstrating radiolucency.

**Figure 5 cancers-14-05275-f005:**
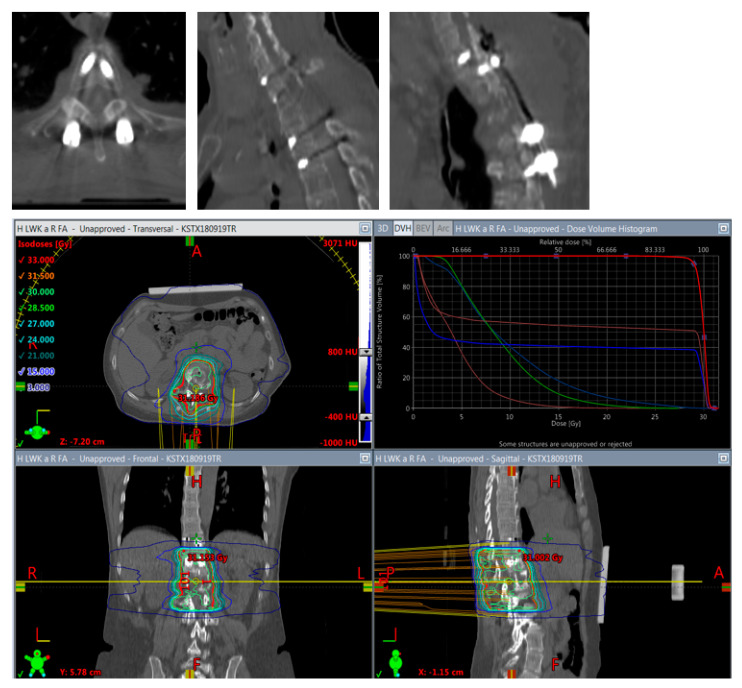
Planning of postoperative radiotherapy: titan. Planning of conventional radiotherapy (30 á 3 Gy) of the vertebral body **plus** all instrumented levels.

**Figure 6 cancers-14-05275-f006:**
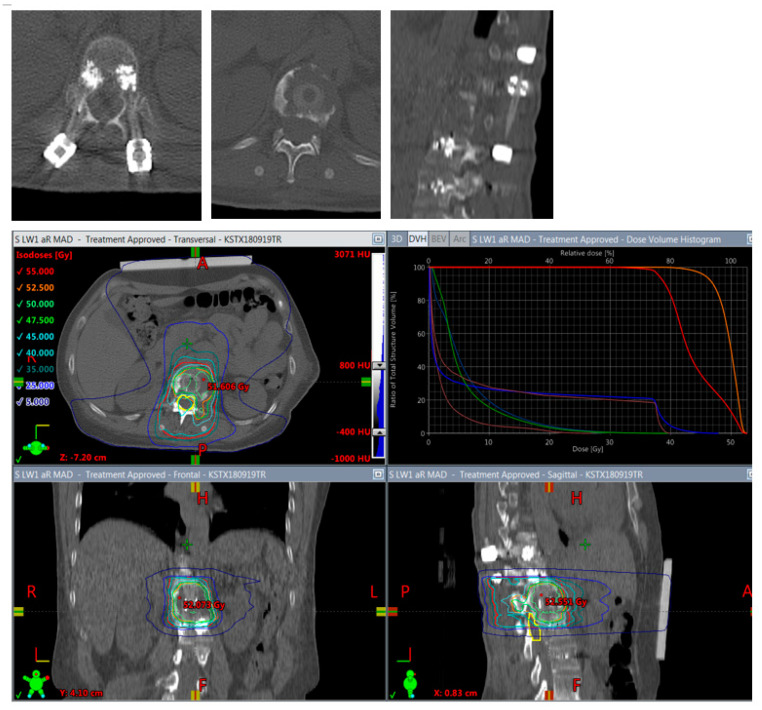
Planning of postoperative radiotherapy: CFR-PEEK. Less artifacts allow smaller volume radiotherapy of 40/50 Gy á 2/2,5 Gy of the vertebral body **alone.** Dose escalation is possible and there is less risk to the spinal cord and others structures at risk.

**Table 1 cancers-14-05275-t001:** Demographic details. Baseline characteristics of patients with spinal metastases and primary bone tumors of the spine are depicted. CUP = cancer of unknown primary, KPS = Karnofsky performance status scale, *n* = number, NSCLC = non-small-cell lung cancer, SD = standard deviation, y = years.

Total, *n*	321
Sex	
Male, *n* (%)	194 (60.4)
Female, *n* (%)	127 (39.6)
Metastases, *n*	306
Primary cancer site, *n* (% of total metastases)	
Prostate	56 (18.3)
Breast	53 (17.3)
NSCLC	43 (14.1)
Kidney	34 (11.1)
CUP	15 (4.9)
other	105 (34.3)
Primary bone tumors, *n*	15
Age, y mean ± SD	65 ± 13
KPS, % mean ± SD	69 ± 19
80–100, *n* (% of total patients)	155 (48.3)
50–70, *n* (% of total patients)	115 (35.8)
0–40, *n* (% of total patients)	51 (15.9)

**Table 2 cancers-14-05275-t002:** Preopeative clinical symptoms. w/o = without.

Preoperative Clinical Symptoms	Number of Patients	Percent of Patients
pain w/o other symptoms	165	51.4%
pain and paresthesia	34	10.6%
incidental finding	20	6.2%
pain and paresis	17	5.3%
pain, paresthesia and paresis	13	4%
ataxia w/o other symptoms	11	3.4%
paresthesia, paresis and bladder/bowel dysfunction	10	3.1%
pain, paresthesia, paresis and bladder/bowel dysfunction	9	2.8%
pain and bladder/bowel dysfunction	9	2.8%
paresis w/o other symptoms	8	2.5%
pain, paresthesia and bladder/bowel dysfunction	7	2.2%
pain, paresis and bladder/bowel dysfunction	5	1.6%
paresis and bladder/bowel dysfunction	5	1.6%
paresthesia w/o other symptoms	4	1.2%
paresthesia and paresis	3	0.9%
paresthesia and bladder/bowel dysfunction	1	0.3%

**Table 3 cancers-14-05275-t003:** Surgical details: MIS = minimally invasive surgery, *n* = number, *n*/a = not available, SD = standard deviation, w/o = without.

	Total *n* = 321
Open approach (*n*)	257 (80.1%)
W/o decompression (*n*)	27
With decompression (*n*)	230
MIS (*n*)	64 (19.9%)
W/o decompression (*n*)	46
With decompression (*n*)	18
Cemented (*n*)	77 (24.0%)
Anterior reconstruction (*n*)	121 (37.7%)
Blood loss (mL) mean ± SD	1104 ± 1146
Red blood cell transfusion	
0	179
1–5	119
>5	14
*n*/a	9

**Table 4 cancers-14-05275-t004:** Intraoperative complications. *n* = number.

	Total (*n* = 321) *n* (%)
Dural tear	15 (4.7%)
Screw breakage	11 (3.4%)
Cement extravasate	4 (1.2%)
Total	30 (9.3%)

**Table 5 cancers-14-05275-t005:** Postoperative complications requiring revision surgery. CSF = cerebrospinal fluid; *n* = number.

	Total (*n* = 321) *n* (%)
Epidural haematoma	6 (1.9%)
Surgical site infection	15 (4.7%)
CSF leackage	7 (2.2%)
Atrophic wound healing disorder	18 (5.6%)
Pedicle screw loosening	7 (2.2%)
Rod breakage (titanium)	1 (0.3%)
Pedicle screw breakage	1 (0.3%)
Total	55 (17.1%)

**Table 6 cancers-14-05275-t006:** Qualitative comparison between pre- and postoperative KPS, and between pre- and postoperative neurological status. KPS = Karnofsky performance status scale, OP = operation.

	Better *n* (%)	Equal *n* (%)	Worse *n* (%)
KPS pre vs. post OP	92 (28.7)	185 (57.6)	44 (13.7)
Neurological status pre vs. post OP	138 (43.0)	134 (41.7)	49 (15.3)

## Data Availability

Data is available in the text.

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
