# Peer review of "CFR-PEEK Pedicle Screw Instrumentation for Spinal Neoplasms: A Single Center Experience on Safety and Efficacy"

_cancers, 2022, doi:10.3390/cancers14215275_

Round 1
Reviewer 1 Report
Thank you for your nice case series.
Can you please consider adding pictures of post operative MRI with the CFR-PEEK screw system in place and maybe an example of a radiosurgery planning screen to show how much better the visualization and accuracy of planning is.
In six cases pedicle screws broke during insertion can you please describe your rescue techniques... did you leave the broken screw in and extend your fusion up a level/ drill out and insert new along same path/ insert new in a different trajectory? I feel this is important to the reader as good fixation is more important at time of first surgery versus revision cases which are often fused or partially fused already
Author Response
Thank you very much for your comments. We added pictures of postoperative MRI and radiotherapy planning (figure 1,2,4,6). We also described the rescue techniques in 3.3.
Reviewer 2 Report
Authors present a single-center retrospective study on use of carbon-fiber-reinforced (CFR) polyethyl-ether-ether-ketone (PEEK) screws in 321 patients with spinal neoplasms (metastases, primary spinal tumors) who underwent posterior stabilization. Intraoperative complications were documented in 30 (9.3%) patients, whereas revision surgery for postoperative complications was necessary in 55 (17.1%) patients. Íntraoperative screw breakage occured in 3.4% and screw loosening in 2.2%.
This study provides important insight into clinical effectiveness and safety of CRF-PEEK screws.
Several suggestions:
1-number of surgeons performing surgeries and relation between experience of the surgeon and postoperative complications / implant failure - were there any screw malpositions;
2- I suggest to remove primary spinal tumors from the total number
3- Question to Figure 1. - when you add the apsolute number of patients written in the Table, you get 306 and not 321 - is there any explanations for this?
4- 30 patients with intraoperative and 55 with postoperative complications; - how many patients in total? Are these 30 patients with intraoperative complications contained into the 55 with postoperative complications (for example, are 7 patients with CSF leak postoperatively among 15 patients who had dural tear?
5-In six cases pedicle screws broke during insertion, in two cases during intraoperative revision and in three cases during implant removal - I suggest to provide more detailed clinical data on this statement - is this the breakage of the pedicle screw itself, or the breakage of the screw out of the pedicle medialy?
6- 80% of patients received radiotherapy? - what is with the rest 20%? Is there a reason why these patients did not receive radiotherapy; low KPS?
7- Is there any comparison for stability and complication rate between CRF-PEEK screws and titanium screws in your institution for patients with spinal metastases, for example comparison to historical cohort prior to introduction of CRF-PEEK screws
8 - For Discussion, I suggest to include and comment:
Burkhardt BW, Bullinger Y, Mueller SJ, Oertel JM. The Surgical Treatment of Pyogenic Spondylodiscitis using Carbon-Fiber-Reinforced Polyether Ether Ketone Implants: Personal Experience of a Series of 81 Consecutive Patients. World Neurosurg. 2021 Jul;151:e495-e506. doi: 10.1016/j.wneu.2021.04.064. Epub 2021 Apr 24. PMID: 33905911.
Krätzig T, Mende KC, Mohme M, Kniep H, Dreimann M, Stangenberg M, Westphal M, Gauer T, Eicker SO. Carbon fiber-reinforced PEEK versus titanium implants: an in vitro comparison of susceptibility artifacts in CT and MR imaging. Neurosurg Rev. 2021 Aug;44(4):2163-2170. doi: 10.1007/s10143-020-01384-2. Epub 2020 Sep 15. PMID: 32930911; PMCID: PMC8338834.
10 - I suggest to include at least one illustrative case with preoperative, intraoperative and postoperative imaging; idealy with postoperative or follow up MRI if applicable
Author Response
1-number of surgeons performing surgeries and relation between experience of the surgeon and postoperative complications / implant failure - were there any screw malpositions;
2- I suggest to remove primary spinal tumors from the total number
3- Question to Figure 1. - when you add the apsolute number of patients written in the Table, you get 306 and not 321 - is there any explanations for this?
4- 30 patients with intraoperative and 55 with postoperative complications; - how many patients in total? Are these 30 patients with intraoperative complications contained into the 55 with postoperative complications (for example, are 7 patients with CSF leak postoperatively among 15 patients who had dural tear?
5-In six cases pedicle screws broke during insertion, in two cases during intraoperative revision and in three cases during implant removal - I suggest to provide more detailed clinical data on this statement - is this the breakage of the pedicle screw itself, or the breakage of the screw out of the pedicle medialy?
6- 80% of patients received radiotherapy? - what is with the rest 20%? Is there a reason why these patients did not receive radiotherapy; low KPS?
7- Is there any comparison for stability and complication rate between CRF-PEEK screws and titanium screws in your institution for patients with spinal metastases, for example comparison to historical cohort prior to introduction of CRF-PEEK screws
8 - For Discussion, I suggest to include and comment:
Burkhardt BW, Bullinger Y, Mueller SJ, Oertel JM. The Surgical Treatment of Pyogenic Spondylodiscitis using Carbon-Fiber-Reinforced Polyether Ether Ketone Implants: Personal Experience of a Series of 81 Consecutive Patients. World Neurosurg. 2021 Jul;151:e495-e506. doi: 10.1016/j.wneu.2021.04.064. Epub 2021 Apr 24. PMID: 33905911.
Krätzig T, Mende KC, Mohme M, Kniep H, Dreimann M, Stangenberg M, Westphal M, Gauer T, Eicker SO. Carbon fiber-reinforced PEEK versus titanium implants: an in vitro comparison of susceptibility artifacts in CT and MR imaging. Neurosurg Rev. 2021 Aug;44(4):2163-2170. doi: 10.1007/s10143-020-01384-2. Epub 2020 Sep 15. PMID: 32930911; PMCID: PMC8338834.
10 - I suggest to include at least one illustrative case with preoperative, intraoperative and postoperative imaging; idealy with postoperative or follow up MRI if applicable
Response:
Thank you very much for your comments:
1) All surgeries were performed by six senior surgeons. We added this to section 2.1. Pedicle screws were inserted using navigation.There was no correlation of implant failure to experience of the surgeon. Intraoperatively, in 39 cases pedicle screws had to be corrected.
2) Thank you very much for this suggestion. Our study focuses on safety and efficacy of CFR-PEEK in clinical practice, and not on the particular oncology of different tumor entities. Especially for primary bone tumors of the spine the advantages of CFR-PEEK, i.e. precise application of postoperative radiotherapy and better visualization of tumor recurrence in follow-up imaging, are of great importance. Hence we prefer to include them to this study.
3) In figure 1 only cases with metastases were depicted. We corrected this. Now this figure shows localization of posterior fusion for metastases and primary bone tumors.
4) 7 patients with an intraoperative complication also had a postoperative complication, so there are in total 78 cases with intra- and postoperative complications. 1 patient with documented intraoperative dural tear had revision surgery for CSF leak, 1 with intraoperative dural tear had revision surgery because of hematoma, 1 with intraoperative dural tear had revision surgery because of screw loosening, 2 with intraoperative screw breakage had revision surgery because of surgical site infection, 1 with intraoperative screw breakage had revision surgery because of hematoma, 1 with intraoperative cement extravasation had revision surgery because of surgical site infection.
5) As CFR-PEEK is less resistant to the torque, screws tend to break during insertion to or removal from osteoblastic bone when higher forces are used. In all our cases with documented screw breakage the bone was osteoblastic. We added more details to section 3.3.
6) This is correct. The 20% of patients who did not receive adjuvant radiotherapy were of low KPS.
7) In a historical cohort of 296 patients operated on between January 1st 2007 and March 31st 2019 because of spinal metastases using titan pedicle screws and rod systems in 21 cases (7.1%) revision surgery was necessary for postoperative complications. A future study to further analyze and compare this is already planned.
8) Thank you for the recommendations. We added these papers to the discussion (section 4).
10) Thank you for this suggestion. We added these images (figure 1, 2, 4).

Round 2
Reviewer 2 Report
The authors have sufficiently responded to reviewer remarks.